# The Association of HHV-6 and the *TNF-α* (-308G/A) Promotor with Major Depressive Disorder Patients and Healthy Controls in Thailand

**DOI:** 10.3390/v15091898

**Published:** 2023-09-08

**Authors:** Sasiwimon Sumala, Tipaya Ekalaksananan, Chamsai Pientong, Surachat Buddhisa, Supaporn Passorn, Sureewan Duangjit, Somwang Janyakhantikul, Areeya Suktus, Sureewan Bumrungthai

**Affiliations:** 1Division of Biotechnology, School of Agriculture and Natural resources, University of Phayao, Phayao 56000, Thailand; 2Department of Microbiology, Faculty of Medicine, Khon Kaen University, Khon Kaen 40002, Thailand; 3HPV & EBV and Carcinogenesis Research Group, Khon Kaen University, Khon Kaen 40002, Thailand; 4Department of Medical Technology, Faculty of Allied Health Sciences, Burapha University, Chonburi 20131, Thailand; 5Division of Pharmaceutical Chemistry and Technology, Faculty of Pharmaceutical Sciences, Ubon Ratchathani University, Ubon Ratchathani 34190, Thailand; 6Division of Biopharmacy, Faculty of Pharmaceutical Sciences, Ubon Ratchathani University, Ubon Ratchathani 34190, Thailand; 7Division of Microbiology and Parasitology, School of Medical Sciences, University of Phayao, Phayao 56000, Thailand

**Keywords:** HHV-6, major depressive disorder (MDD), *TNF-α*, inflammation

## Abstract

Major depressive disorder (MDD) is a silent global health problem that can lead to suicide. MDD development is suggested to result from numerous risk factors, including genetic factors. A precise tool for MDD diagnosis is currently not available. Recently, inflammatory processes have been identified as being strongly involved in MDD development and the reactivation of human herpesvirus type 6 (HHV-6), upregulating cytokines such as TNF-*α*, which are associated with MDD. Therefore, this study aimed to determine the association of HHV-6 with genetic factors, especially *TNF-α* mutation, in MDD patients and their relatives compared to healthy controls. The Patient Health Questionnaire (PHQ-9) was used to evaluate MDD status, and 471 oral buccal samples were investigated for HHV-6 infection and viral copy number by qPCR. *TNF-α* (-308G/A) gene mutation and the cytokines TNF-α, IL-6, and IL-10 were analyzed by high-resolution melting (HRM) analysis and enzyme-linked immunosorbent assay (ELISA). Whole-exome sequencing of buccal samples was performed to analyze for genetic factors. The results showed significantly higher HHV-6 positivities and viral loads in MDD patients (15/59 (25.67%) and 14,473 ± 16,948 copies/µL DNA) and their relatives (blood relatives 17/36 (47.22%) and 8146 ± 5656 copies/µL DNA); non-blood relatives 7/16 (43.75%) and 20,721 ± 12,458 copies/µL DNA) compared to the healthy population (51/360 (14.17%) and 6303 ± 5791 copies/µL DNA). The *TNF-α* (-308G/A) mutation showed no significant difference. Surprisingly, 12/26 (46.15%) participants with the *TNF-α* (-308G/A) mutation showed HHV-6 positivities at higher rates than those with wild-type *TNF-α* (-308G) (70/267 (26.22%)). HHV-6-positive participants with *TNF-α* (-308G/A) showed higher levels of TNF-α, IL-6, and IL-10 than those of negative control. Exome analysis revealed that common mutations in immune genes were associated with depression. Therefore, this study unveiled the novel association of inflammatory gene *TNF-α* (-308G/A) mutations with HHV-6 reactivation, which could represent a combined risk factor for MDD. This result could induce further research on MDD development and clinical applications.

## 1. Introduction

Major depressive disorder (MDD) is a silent global health problem that can result in suicide. According to the World Health Organization, 280 million people (approximately 3.8–13.0% [1,2,3] of the global population) are afflicted with MDD [1], and in Thailand, the MDD prevalence was 7.0–39.1%, which may relate to the lack of knowledge regarding depression and the absence of precise diagnostic tools for this condition [1,4,5,6,7,8,9,10,11]. The Thai version of the Patient Health Questionnaire (PHQ-9) for MDD screening is the main tool available, which is a nine-item questionnaire based on the Diagnostic and Statistical Manual of Mental Disorders, Fourth Edition (DSM-IV) criteria that are used for diagnosing major depressive episodes [1,4,5,6,7,8,9,10,11]. However, accurate MDD diagnosis still requires a precise tool or gold standard biomarker.

Many risk factors are suggested to be associated with the development of MDD, and meta-analyses have identified several environmental risk factors including the loss of a spouse, physical abuse during childhood, obesity, the presence of 4–5 metabolic risk factors, sexual dysfunction, and work-related stress [12]. Other associated risk factors include high body mass index (BMI), smoking [13], female sex [14,15], family and personal psychiatric history, family history of mental disorders, and alcohol consumption [16]. In addition, genetic risk factors [12] such as genetic polymorphism of *tumor necrosis factor-alpha* (*TNF-α*) (-308G/A; rs1800629) [17,18,19,20,21], *interleukin* (*IL)-1β* (rs1143627) [22], *IL-6* (174G/C; rs1800795) [23], *5HTTP/SLC6A4* (rs25531), *APOE*, *DRD4*, *GNB3*, *HTR1A*, *MTHFR*, and *SLC6A3* have been significantly associated with MDD [24,25]. Furthermore, numerous studies have found that families and twins had increased risks of MDD [16,26], whereby the heritability of MDD in twins is estimated at 38% [27], and the odds ratio in first-degree relatives was determined to be 2.84 in a meta-analysis [28]. In contrast, a meta-analysis of genetic variation showed that the proportion of variance explained by genetic risk factors was extremely small (0.1–0.4%) [12], possibly because other indirect or direct factors were involved that related to MDD in close relatives (but not twins), such as infectious particles; however, the reason for the differences in the results of these studies has yet to be determined.

Human herpesvirus type 6 (HHV-6) is a 160–162 kilobase (kb) double-stranded linear DNA virus that belongs to the family *Herpesviridae* and includes the HHV-6B and HHV-6A variants. Adults in developed countries show a greater than 95% seropositivity for HHV-6 [29,30,31]. Latent infections of HHV-6 in astrocytoma cell lines have resulted in the production of IL-6 [32], and a high level of IL-6 is associated with increased suicide risk. Antidepressant treatment has been shown to significantly decrease peripheral levels of IL-6, TNF-α, and IL-10 [13,33], while IL-6, TNF-α, IL-10, IL-13, IL-18, and IL-12 are associated with MDD severity level [34]. In contrast, the levels of IL-1β, IL-2, IL-4, IL-8, soluble IL-6 receptor (sIL-6R), IL-5, Chemokine (C-C motif) ligand 3 (CCL-3), IL-17, and transforming growth factor beta (TGF-β 1) were not considered significant in relation to MDD. Inflammatory processes involving cytokine production are reported to be associated with MDD, including neurotrophic viral infections, and since HHV-6 infection reflects an inflammatory response, it has an affiliation with MDD [34,35,36,37,38,39,40]. Transmission of HHV-6 mainly occurs through saliva, nasal mucus, and the olfactory bulb which guides the axons of olfactory receptor neurons into the CNS in healthy populations. The genome of latent HHV-6 integrates into the telomeres of the host chromosome. Active infection of HHV-6 in MDD patients has been predominantly detected within the Purkinje cells of the cerebellum. A small protein encoded by an intermediate state transcript (SITH-1) is produced through the expression of an HHV-6 latency-associated gene, which induces depression symptoms. The corticotropin-releasing hormone (CRH) along with the urocortin, and REDD1 proteins are activated by HHV-6B latency in the hypothalamus [37,38,39].

The “cytokine theory” of MDD explains the various behavioral, neuroendocrinal, and neurochemical changes involved in depression [41]. Although some studies have reported that peripheral cytokines do not cross the blood–brain barrier, molecular, cellular, and neural signals have been identified that activate brain inflammation [13]. However, immune cells have been recognized to infiltrate nervous tissue and cytokine signals can be transmitted to the nervous system. Moreover, the CNS (astrocytes, microglia, and in some cases, neurons) can generate cytokines and their receptors [42]. HHV-6 reactivates and persists in the host by avoiding immune surveillance through methods such as the immunosuppression of CD4^+^ and CD8^+^ T cells, B cells, monocytes/macrophages, and NK cells [43]. HHV-6 increases specific pro-inflammatory gene expression and downregulates anti-inflammatory genes [44,45]. IFN-γ release was found to be inhibited by HHV-6 in peripheral blood mononuclear cells (PBMCs) [43,46,47]. In addition, HHV-6 induced TNF-α upregulation in PBMCs [48] and differentiated U937 monocytoma cells [49]. Furthermore, the HHV-6 reactivation of CD4^+^ continuous JJHAN T cells increased TNF receptor 1 expression, resulting in the apoptosis of CD4^+^ lymphocytes from patients in vivo [50]. IL-10 and IL-14 were also found to be downregulated by HHV-6 infection in continuous T cells [1,47]. However, HHV-6 induced helper T cell (Th) differentiation from the Th1 to the Th2 type by upregulating IL-10 and downregulating IL-12 in PBMCs because of monocyte/macrophage-induced IL-10 expression [43]. IL-18 was increased in T cells infected with HHV-6 and the levels of IL-10 were reduced, whereas IL-12 production in stimulated macrophages was deficient after HHV-6 infection. HHV-6 has been reported to induce IL-6 both in vitro and in vivo, and upregulate the production of monocyte chemoattractant protein-1 (MCP-1) and IL-8 [47,51]. IL-6 levels approximating 0–175 pg/mL were detected in HHV-6-positive cells [52] and were undetected in uninfected cells [47]. CD3 has been shown to be downregulated by HHV-6, resulting in reduced surface expression of the CD3/T cell receptor complex. Similarly, T cell proliferation was severely impaired after HHV-6 infection [50]. HHV-6 can frequently lead to acute inflammation, but the underlying mechanisms remain unclear [45].

The presence of anti-HHV-6 IgG in serum can be detected with high sensitivity and specificity [34], and reactivation can be identified in approximately 1.75–2% of patients 3 years of age or younger by polymerase chain reaction (PCR) [53,54]. Using this method, the global spread of HHV-6 was determined as 28–78% [55,56]. Currently, sensitive methods for the detection of HHV-6 in saliva are available [57]. Buccal mucosa swabs and whole blood samples were used in the Japanese population to determine the HHV-6 copy number [53,58]. Nevertheless, depending on viral neurotropism, negative results in blood or saliva samples by serology or PCR cannot guarantee HHV-6 negativity in the brain [34]. HHV-6 might be a risk factor for MDD; therefore, an MDD diagnosis should not be based solely on a questionnaire. HHV-6 may be an additional factor to environmental risks that stimulates and increases the risk of recurrent inflammatory processes in MDD.

Investigations that highlight the correlation between the genetic markers of *TNF-α* gene variations, which are the most common genetic biomarkers, and MDD in the Thai population are currently scarce. Therefore, the objective of this study was to determine the significance of *TNF-α* mutation in MDD and the association between HHV-6 reactivation and MDD in Thailand. The presence of HHV-6 DNA was detected using quantitative PCR (qPCR) with oral buccal cells from MDD patients, relatives of MDD patients, and a healthy population, and the HHV-6 viral load was determined. The *TNF-α* (-308G/A; rs1800629) mutation was identified by high-resolution melting (HRM) analysis, and the cytokine level of TNF-α with IL-6 and IL-10 was detected by enzyme-linked immunosorbent assay (ELISA). Whole-exome sequencing (WES) was used to compare patients with MDD, relatives of patients with MDD, and healthy individuals to screen for genetic risk factors related to the Thai population.

## 2. Materials and Methods

### 2.1. Specimens

Total of 471 oral buccal cell samples were collected from 360 healthy individuals, 59 MDD patients, 36 blood relatives of MDD patients, and 16 non-blood relatives of MDD patients (such as spouses or partners) for HHV-6 and *TNF-α* promoter (-308G/A; rs1800629) detection (Figure 1). The sample size was calculated as N = Z^2^_1−a_ P(1 − P)/d^2^ for an HHV-6 prevalence (P) of approximately 4–53.8 %, with Z = 1.96 for a 95 % confidence level and d = 0.05 [59,60]. The inclusion criteria used for patients who were diagnosed with MDD were the DMS-5 criteria from the Thai population between 2022–2023 for patients aged 18 to 90 years. The inclusion criterion for the healthy controls was the absence of a current or lifetime diagnosis of any psychiatric disorder. First-degree relatives with a history of suicide risk or depression diagnoses were excluded from participating as healthy controls. MDD patients were diagnosed by a certified hospital psychiatrist. This study evaluated the potential risk factors of MDD in families of MDD patients; therefore, the study was promoted using online social media among depression patients in Thailand. Oral buccal cell samples were collected using mail-in samples and medical certifications were used to confirm the MDD diagnosis with the PHQ-9. The results were interpreted as no depression (0–7), mild depression (8–12), moderate depression (13–17), and major depression (18 or greater). A PHQ-9 score of ≥9 was used to identify patients with depression, according to a previous study [61,62].

A randomized sample matching the sex of 60 healthy individuals and 59 MDD patients was used to confirm the most common risk factors of MDD. Data were collected regarding sex, health status, and life history, BMI, exercise habits, alcohol consumption, second-hand smoke exposure, drinking water source, potable water purification, water type used for brushing teeth, and fresh fruit and vegetable consumption.

An ELISA was performed using 20 randomized undiluted serum samples, including those from healthy participants and MDD patients with known HHV-6/*TNF-α* promoter status.

Whole-exome sequencing of buccal samples was used to analyze the point mutation status of four randomized samples (an MDD patient and a healthy relative with a healthy subject).

This study was approved by the Committee on Human Research Ethics in Health Sciences and Science and Technology, University of Phayao, Thailand (UP-HEC 1.3/013/65). All procedures involving human participants performed in the study were in accordance with the ethical standards of the Declaration of Helsinki, the Belmont Report, the Council for International Organizations of Medical Sciences guidelines, and the International Conference on Harmonization in Good Clinical Practice.

### 2.2. DNA Extraction

Oral buccal cells were collected by oral rinsing with phosphate-buffered saline (PBS). Samples were centrifuged at 4000× *g* for 5 min to remove the supernatant. DNA was extracted from the oral buccal cells using the Genomic DNA Isolation Kit (PDC11-0100; BIO-HELIX Co., Xindian District, New Taipei City, Taiwan, China) according to the methods of a previous study [63].

### 2.3. HHV-6 DNA Detection by Quantitative PCR (qPCR) and Viral Load

HHV-6 DNA (*U97* gene) was detected using qPCR with the forward primer 5′ GCTAGAACGTATTTGCTGCAGAACG 3′ and the reverse primer 5′ ATCCGAAACAACTGTCTGACTGGCA 3′, and a PCR product size of 259 base pairs (bp). An in-house plasmid containing the HHV-6 *U97* gene (259 bp) was used as the positive control. The experiment was performed in duplicate. The PCR reaction was conducted using 5 × FiREPOL Eva Green qPCR Mix Plus (Solis Bio Dyne, Tartu, Estonia) according to the protocol of a previous study [63]. The *B-globin* gene was used as an internal control and was detected using the GH_2_O forward primer 5′ GAAGAGCCAAGGACAGGTAC 3′ and the PCO_4_ reverse primer 5′ CAACTTCATCCACGTTCACC 3′, with a PCR product size of 268 bp [64].

The HHV-6 positive cases were confirmed for HHV-6 viral load in duplicate. The positive controls were diluted 10-fold in duplicate to create a standard curve. The number of copies was determined using the following formula:
Number of copies =

 (Z ng × 6.0221 × 10^23^ molecules/mole)/(N × 660 g/mole) × 1 × 10^9^ ng/g
where Z = amount of amplicon (ng), N = length of dsDNA amplicon, 660 g/mol = average mass of 1 bp dsDNA, 6.022 × 10^23^ = Avogadro’s constant, and 1 × 10^9^ = conversion factor.

### 2.4. TNF-α Promoter (-308G/A; rs1800629) Detection by High-Resolution Melt (HRM) Analysis

*TNF-α* (-308G/A; rs1800629) was detected by HRM using the forward primer 5′CACAGACCTGGTCCCCAAAA 3′ and the reverse primer 5′ CATCCTCCCTGCTCCGATTC 3′, with a PCR product size of 136 bp. The experiment was performed in duplicate. The PCR reaction was performed using 5 × FiREPOL Eva Green HRM Mix Plus (Solis Bio Dyne, Tartu, Estonia) according to a previous study [63]. DNA samples known to have *TNF-α* with -308G/A and -308G were used as positive controls.

### 2.5. DNA Sequencing

Sanger sequencing was used to confirm the *TNF-α* promoter point mutation (-308G/A). The *TNF-α* primers identified in Section 2.4 were used as the forward and reward sequences for PCR DNA sequencing. The sequences were analyzed by comparison to reference sequences (>ref|NC_000006.12|:31,575,208–31,575,322 *Homo sapiens* chromosome 6, GRCh38.p13 Primary Assembly ACCTGGTCCCCAAAAGAAATGGAGGCAATAGGTTTTGAGGGGCATGGGGACGGGGTTCAGCCTCCAGGGTCCTACACACAAATCAGTCAGTGGCCCAGAAGACCCCCCTCGGAAT) from GenBank using Bioedit version 7.2 (a biological sequence alignment editor).

### 2.6. Screening Effect of HHV-6 Status and TNF-α Promoter (-308G/A) Mutation by ELISA

An ELISA was performed on 20 samples including healthy participants and MDD patients with known HHV-6 (positive or negative)/*TNF-α* promoter status (healthy: positive/G *n* = 3, positive/G/A *n* = 5, negative/G/A *n* = 5, negative/G *n* = 5; MDD patients treated with drugs: positive/G *n* = 1, negative/G *n* = 1). The serum levels of pro-inflammatory cytokines (IL-6, TNF-α) and anti-inflammatory cytokines (IL-10) were measured using a human cytokine IL-6, IL-10, and TNF-α ELISA kit (BD Biosciences, San Jose, CA, USA) according to the manufacturer’s instructions and the methods of a previous study [65]. The experiment was performed in duplicate or triplicate.

### 2.7. Whole-Exome Sequencing (WES)

Whole-exome sequencing was used to analyze the point mutation status of samples from four participants: an MDD patient (female), the first-degree relative of an MDD patient (male; brother), a healthy female, and a healthy male aged between 21–30 years. DNA was extracted using previously described methods and instructions, and DNA was eluted using TE buffer. Whole-exome sequencing was conducted with no replications. DNA quality control, library preparation, library quality control, cluster generation, and sequencing were performed. The precision and stability of each experiment were necessary to ensure the reliability of the subsequent bioinformatics analysis. A qualifying DNA sample was fragmented using an ultra-sonicator and a resultant fragment was constructed into a high-throughput sequencing library through the steps of terminal repair, base A tail addition, adaptor ligation, purification and pre-amplification, quantitation, exon capture, and PCR enrichment. After the completion of library preparation, the size and concentration of each sample were determined, and a Qubit fluorometer was used for the accurate measurement of DNA concentration. The quality and concentration of the sequencing library were assessed.

### 2.8. Statistical Analysis

The data were analyzed using IBM SPSS software version 16, and *p* < 0.05 was considered statistically significant. Pearson’s chi-squared or Fisher’s exact test was used to compare the categorical variables between groups. An independent Student’s *t*-test or unpaired *t*-test was used to compare separate mean ± standard deviation (SD) sets. A one-way ANOVA, Median test, Mann–Whitney U-test, or Kruskal–Wallis test was used to compare the mean/median ± SD or SEM of >3 groups.

## 3. Results

### 3.1. Risk Factor for MDD

Online samples were collected and confirmed as MDD based on medical certifications and the PHQ-9. A total of 60 healthy participants and 59 patients with depression were studied. Of the MDD patients, 89% reported mild-to-major depression based on the PHQ-9, while 93% of healthy participants disclosed no-to-mild depression.

Education, income, familial relationships, BMI, exercise, fermented food consumption, water type used for brushing teeth, and second-hand smoke exposure were associated with MDD (Table 1).

### 3.2. HHV-6 Detection by qPCR

Oral buccal cells were collected from 471 donors from the Thai population and HHV-6 was detected using qPCR. The results showed a statistically significant difference in HHV-6 positivity between patients with depression (15/59 (25.67%)) and health participants (51/360 (14.17%)); *p*-value = 0.028 (odds ratio = 2.066, 95% CI = 1.071–3.983). The results are listed in Table 2, which also details the high prevalence of HHV-6-infected blood relatives and non-blood relatives of MDD patients.

This study determined the relationships among HHV-6 infection in MDD patients, blood relatives of MDD patients, and non-blood relatives of MDD patients, as shown in Figure 2.

No statistically significant difference was observed between MDD patients, blood relatives of MDD patients (17/36 (47.20%)), and non-blood relatives of MDD patients (7/16 (43.75%); *p*-value = 0.076). Moreover, the healthy population included a lower number of HHV-6-positive cases than the healthy blood relatives and non-blood relatives of MDD patients (*p*-value = 0.000 and 0.001, respectively). Thus, a higher prevalence of HHV-6 infection was found among the family members (healthy) of MDD patients than the healthy control population.

Education, income, familial relationships, BMI, exercise, water type used for brushing teeth, and second-hand smoke exposure were associated in all MDD (59 cases) and MDD without HHV-6 cases. Only familial relationships and exercise were associated with MDD with HHV-6 positivity, as shown in Figure 3 and Appendix A.

Figure 4 shows the PHQ-9 level of the healthy individual controls and MDD patients in HHV-6 positive and negative cases. HHV-6 positive cases had a higher percentage of major depression based on the PHQ-9 level (highest severity level) than the HHV-6 negative level in MDD patients and healthy individual controls; however, the difference was not statistically significant. The result might be error because of PHQ-9 score ≥ 10 had a sensitivity of 88% and a specificity of 88% for major depression.

### 3.3. TNF-α (-308G/A; rs1800629) Promoter Detection by HRM Analysis

The *TNF-α* promoter (-308G/A) was detected using HRM. The results showed no statistically significant difference between the presence of G/A in depressed patients (3/39 (7.69%)) and that in the healthy population (15/220 (6.8%)); *p*-value = 0.843 (odds ratio = 1.139, 95% CI = 0.314–4.134). No statistically significant difference was observed between MDD patients and blood relatives (6/20 (30%)) and between MDD patients and non-blood relatives (2/14 (14.29%)) (*p*-value = 0.072). The results are shown in Table 3.

No statistically significant difference was observed between the healthy population and non-blood relatives of MDD patients (*p*-value = 0.279).

Surprisingly, a relationship was noted between the *TNF-α* (-308G/A; rs1800629) promoter mutation and HHV-6 infection (Figure 5).

Individuals with the *TNF-α* (-308G/A) promoter mutation showed a higher frequency of HHV-6 infection (12/26 (46.15%)) than those with the wild-type *TNF-α* (-308G) (70/267 (26.22%)); *p*-value = 0.031 (odds ratio = 2.412, 95% CI = 1.065–5.465). Thus, the *TNF-α* (-308G/A) promoter mutation could affect HHV-6 infection. The prevalence of HHV-6 infection, the *TNF-α* (-308G/A) promotor mutation, and MDD patients in Asia are shown in Table 4.

### 3.4. HHV-6 Viral Load

qPCR was used for viral load detection in the 90 HHV-6 positive cases. The results showed a statistically significantly higher load of HHV-6 in MDD patients (14,473 ± 16,948 copies/ng DNA) than in the healthy population (6303 ± 5791 copies/ng DNA); *p*-value = 0.044 (Figure 6).

MDD patients had HHV-6 viral loads similar to those of their non-blood relatives (20,721 ± 12,458 copies/ng DNA) and blood relatives (8146 ± 5656 copies/ng DNA; *p*-value 0.052). The healthy population had HHV-6 viral loads similar to those of blood relatives of MDD patients (*p*-value = 0.093). The viral load of HHV-6 was found to be higher among family members (healthy) of MDD patients than among healthy controls.

Based on the PHQ-9 level of the participants, the viral load was 2266 copies/ng DNA in those with no depression, 26,010 copies/ng DNA in participants with mild depression, 8557 copies/µL DNA in people with moderate depression, and 16,072 copies/µL DNA in those with major depression.

This study did not find a relationship between the *TNF-α* (-308G/A) promoter mutation and HHV-6 viral load. Samples with the *TNF-α* (-308G/A) promoter mutation had HHV-6 viral loads similar to those with the wild-type *TNF-α* (-308G) (8148 ± 5120 copies/ng DNA and 9842 ± 11,235 copies/ng DNA, respectively; *p*-value = 0.422).

### 3.5. ELISA

Twenty samples from normal and MDD patients with various HHV-6 (positive or negative)/*TNF-α* promoter statuses (healthy: positive/G *n* = 3, positive/G/A *n* = 5, negative/G/A *n* = 5, and negative/G *n* = 5; MDD patients being treated with drugs: positive/G *n* = 1 and negative/G *n* = 1) were available for the ELISA.

The results showed that the levels of IL-6, IL-10 were statistically significantly different (*p*-value = 0.002 and 0.013, respectively) when using the Median test.

The results showed that the levels of TNF-α was not statistically significantly different (*p*-value = 0.441) when using the Median test.

Surprising, the results showed that the levels of IL-6, IL-10, and TNF-α were higher in healthy participants with HHV-6 positive/G/A status than in healthy participants with HHV-6 negative/G/A status (*p*-value = 0.004, 0.038, and 0.042, respectively, using the Kruskal–Wallis or unpaired *t*-test). The results are shown in Figure 6.

*TNF-α* mutations (-308G/A) are a risk factor of MDD and the results showed that the *TNF-α* mutation (-308G/A) induced high expression of TNF-α with the IL-6 and IL-10 signaling pathway in HHV-6 positive individuals.

HHV-6 might have affected the observed association with *TNF-α* gene variation in this study, considering its influence on TNF-α cytokine levels.

### 3.6. Whole-Exome Sequencing

The results of whole-exome sequencing showed the olfactory receptor family 8 subfamily G member 2 (OR8G2P; rs141204263) and olfactory receptor family 2 subfamily T member 4 (OR2T4) in MDD patients but not in healthy family members. Olfactory receptors are chemoreceptors expressed in the cell membranes of olfactory receptor neurons.

This study did not find genes commonly associated with MDD, such as serotonin, dopamine or *SLC6A4*, *GNAZ*, *DRD2*, *MMP-1*, *SHANK3*, *VDR*, *GRIA1*, *ANKK1*, *TPH1*, and *RNF180* mutations in the MDD patients. Genes associated with depression or neuro-diseases were found in this study, such as *ABCG1*, *ASMTL*, *CACNA1F*, *COX7A1*, *GRID2*, *HTR3E*, and *SHANK2* (Table 5).

In addition, this study found gene mutations associated with the immune system, such as complement, interferon, interleukin, scavenger receptor, Toll-like receptor, TNF receptor, and CD molecule (Figure 7).

## 4. Discussion

This study identified that HHV-6 has implications on the observed association between *TNF-α* gene variation and MDD. This influence on *TNF-α* mutation could have an impact on the immunological homeostasis of viral reactivation. Currently, information regarding the prevalence of the -308G/A allele in people with depression and its relationship with HHV-6 reactivation and viral load in terms of understanding MDD’s complex etiology is limited. Table 4 shows trend of HHV-6 prevalence and MDD in Asia (Appendix A).

The detection of HHV-6 in patients diagnosed with encephalitis or meningitis using PCR strongly indicates that HHV-6 can affect central nervous system diseases [53,123]. An autopsy brain sample of a patient was positive for HHV-6 and showed a high viral load; however, a low level of HHV-6 infection was found in the cerebrospinal fluid (CSF) [124]. In Japanese patients, HHV-6 infection is a common cause of acute sporadic encephalopathy but cannot be found in the CSF. These results indicate that HHV-6 encephalopathy is caused by an indirect mechanism [52]. Finally, although the biomarker might not consist of viral particles, the present study showed a new association of HHV-6 that could induce the severity of MDD, although, no statistical significant difference between HHV-6 positive cases with higher percentages at the PHQ-9 level (major depression) and HHV-6 negative cases were found. Increasing MDD sample sizes will be required to confirm the results of this experiment in further studies.

Genetic variations are risk factors for MDD, although meta-analyses of environmental risk factors have shown that the proportion of variance explained by genetic risk factors is extremely small (0.1–0.4%) [12]. A systematic review of *TNF-α* (-308G/A) showed both effects and an absence of effects in relation to MDD [49]. The present study suggested that *TNF-α* (-308G/A) was not significantly found in MDD and healthy individuals; however, the association between *TNF-α* (-308G/A) and HHV-6 reactivation was prevalent in MDD cases. A systematic review and meta-analysis showed that TNF-*α*, IL-6, and IL-10 levels were associated with MDD [33]. IL-6 production is typically preceded by the release of TNF-α, triggering inflammation. The current study suggested that both *TNF-α* (-308G/A) and HHV-6 might affect TNF-α, IL-6, and IL-10 expression, influencing the severity and progression of depression via inflammatory processes. Only familial relationships and exercise were associated with MDD cases that were HHV-6 positive as opposed to MDD cases without HHV-6. Several meta-analyses found that exercise can decrease circulating inflammatory factors such as IL-6, IL-18, C-reactive protein (CRP), leptin, fibrinogen, and angiotensin II to create an anti-inflammatory environment [125,126]. HHV-6 cannot be eradicated throughout the lifetime of the host. However, exercise appears to reduce the effect of HHV-6 on inflammation. This result suggests that the mechanism of HHV-6 positivity that induces MDD might be different from that of uninfected MDD. Based on a meta-analysis, Köhler et al. [12] found that family problems and exercise [125,126] were highly significantly associated with MDD.

Inflammation is associated with MDD [40] and a chronic inflammatory state involves increased activation of microglia, IL-1β, IL-6, and TNF-α. The persistent activation of microglia leads to the inefficient clearance of neurotoxic molecules, as well as neuron loss and a reduction in neurogenesis. Cytokines induce indoleamine 2,3-dioxygenase, an enzyme that reduces serotonin production [14,127], which causes decreased neurogenesis and a reduced number and size of glial cells in the hippocampus [128,129]. The presence of IL-1, IL-6, and TNF-α indicates resistance in the modulation of glucocorticoids, maintenance of the increased activity of the HPA axis [130], and affectation of neurogenesis in the hippocampus due to the reduction of brain-derived neurotrophic factor. Increased levels of TNF-α were found in the dorsal–anterior cingulate cortex and the anterior insula of postmortem patients who presented with social rejection and increased anxiety [131]. Drugs that affect serotonin mechanisms could reduce inflammation and thereby reduce the effect of HHV-6. However, no mechanisms to remove HHV-6 from the brain, blood, or oral cells of infected patients have been identified. HHV-6 might affect the recurrence of MDD in patients after medication is stopped, due to stress or hypochondria because of a doubly activated inflammatory process.

Cytokine expression could be a potential biomarker of MDD. Antidepressant treatment significantly decreased the peripheral levels of IL-6, TNF-α, and IL-10 [13], and new evidence shows that anti-inflammatory drugs might be useful as antidepressants in MDD patients, such as in relation to IL-6 [14].

Exome sequencing is a genomic technique for sequencing gene mutations. This study found gene mutations associated with immune system components such as complement, interferon, interleukin, scavenger receptor, Toll-like receptor, TNF receptor, and CD molecule in MDD patients. These results are similar to those of Ning et al. [132], who determined that CD1C could be a useful biomarker for future studies in Thailand.

A limitation of this study was the low sample size of MDD patients and their relatives. Further studies should increase the sample size to improve the power of statistics. This study had bias as the mean ages in both groups were not a match and the result should be considered. The *TNF-α* mutation with cytokine level result showed a low power of statistical analysis. Therefore, to confirm this result, a larger group with the MDD *TNF-α* (-308G/A) mutation is required. No data are available on disease treatment or the exclusion of other chronic or acute inflammatory states. No conclusions can be drawn from the next-generation sequencing analysis since only four individuals were analyzed. However, a high prevalence of HHV-6 infection in the population and its ability to reactivate under any inflammatory or immunosuppressive conditions makes it difficult to discern whether an association between HHV-6 reactivation and a disease is causal or epiphenomenal and must be confirmed in the future.

## 5. Conclusions

Overall, this study revealed that HHV-6 infection is associated with MDD patients in the Thai population. *TNF-α* (-308G/A) gene mutations with HHV-6 positivity showed higher expression of TNF-α with the IL-6 and IL-10 signaling pathway than those with HHV-6 negativity. Novel insights derived from this study regarding *TNF-α* (-308G/A) mutations, HHV-6 and MDD could present a basis for future research and clinical applications. The relevance of investigating *TNF-α* gene variation and HHV-6 infection in the context of MDD could provide a better understanding of this association among the Thai population.

## Figures and Tables

**Figure 1 viruses-15-01898-f001:**
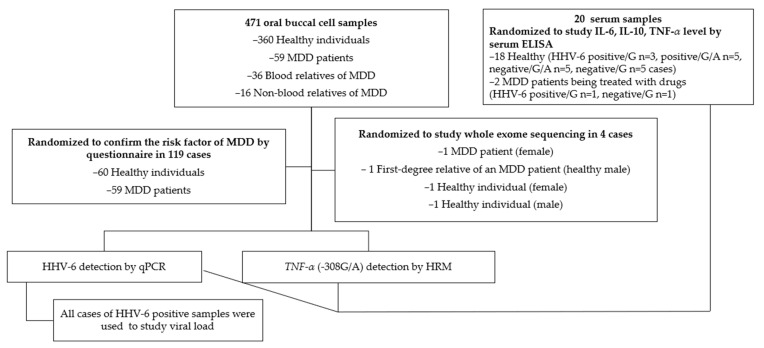
Specimens.

**Figure 2 viruses-15-01898-f002:**
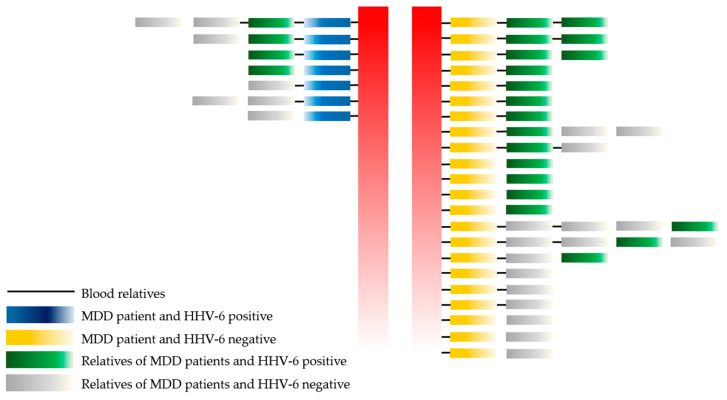
The association of HHV-6 with MDD patients, blood relatives, and non-blood relatives. Remark: red color = separate between HHV-6 positive and negative in MDD group.

**Figure 3 viruses-15-01898-f003:**
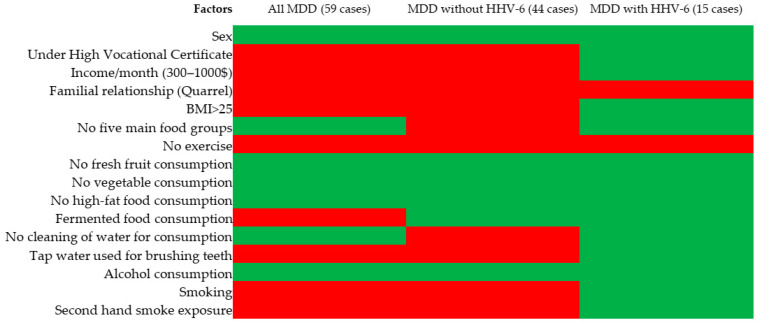
Risk factors for MDD, MDD without HHV-6, MDD with HHV-6. Remark: red = statistically significant, green = not statistically significant when associated with MDD compared to healthy control.

**Figure 4 viruses-15-01898-f004:**
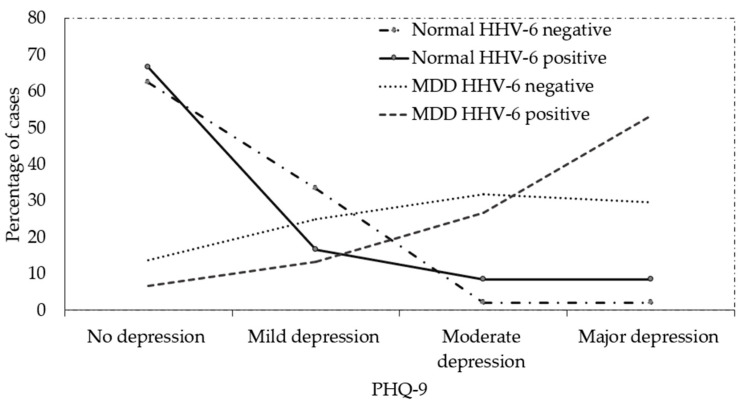
PHQ-9 level of healthy individual control and MDD patients in HHV-6 positive and negative cases (percentage).

**Figure 5 viruses-15-01898-f005:**
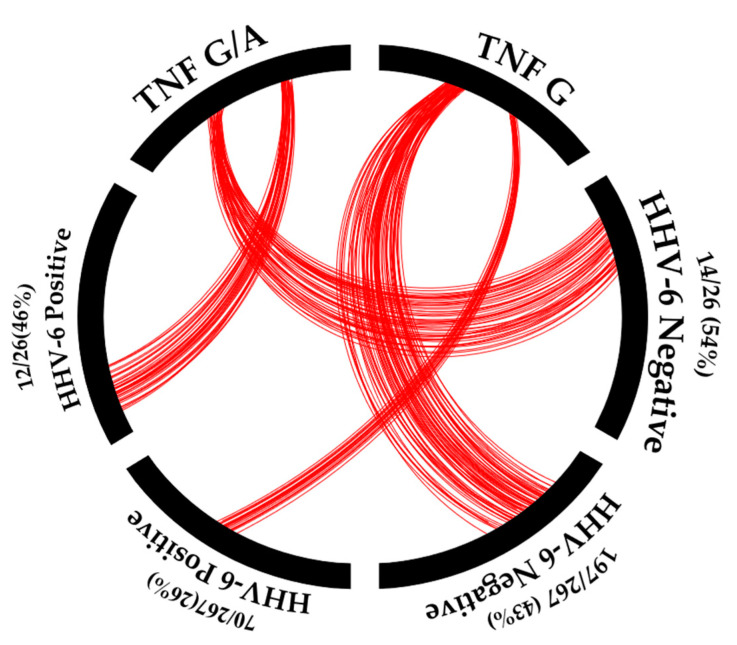
The association of HHV-6 positivity with *TNF-α* promotor status. Remark: red = number of samples.

**Figure 6 viruses-15-01898-f006:**
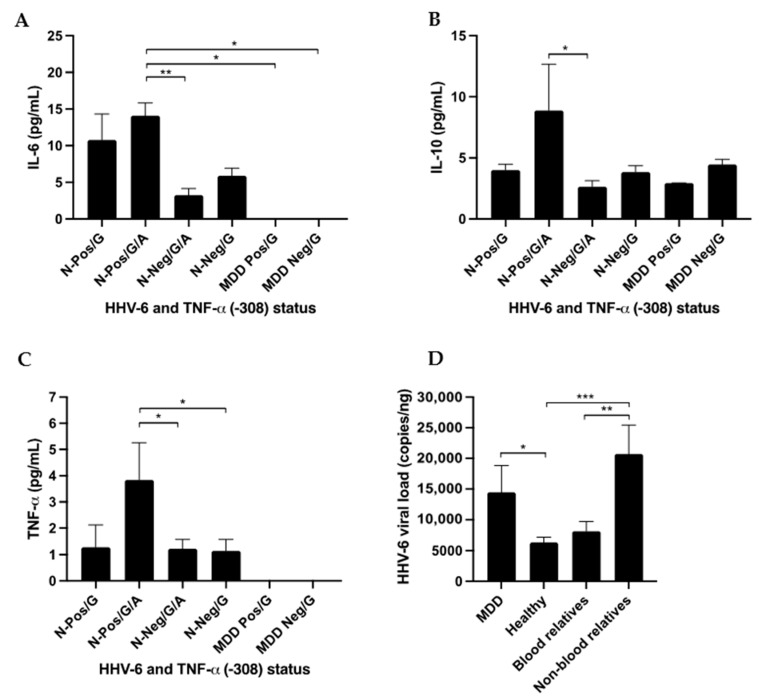
The association of HHV-6 and *TNF-α* promotor status with cytokine expression measured using ELISA and HHV-6 viral load. N = normal healthy control. Healthy: positive/G, positive/G/A, negative/G/A, negative/G; and MDD patients being treated with drugs: positive/G, negative/G were available for *ELISA* analysis. (**A**) IL-6 cytokine level, (**B**) IL-10 cytokine level, (**C**) TNF-α cytokine level, and (**D**) HHV-6 viral load. * *p*-value < 0.05, ** *p*-value < 0.01 and *** *p*-value < 0.001.

**Figure 7 viruses-15-01898-f007:**
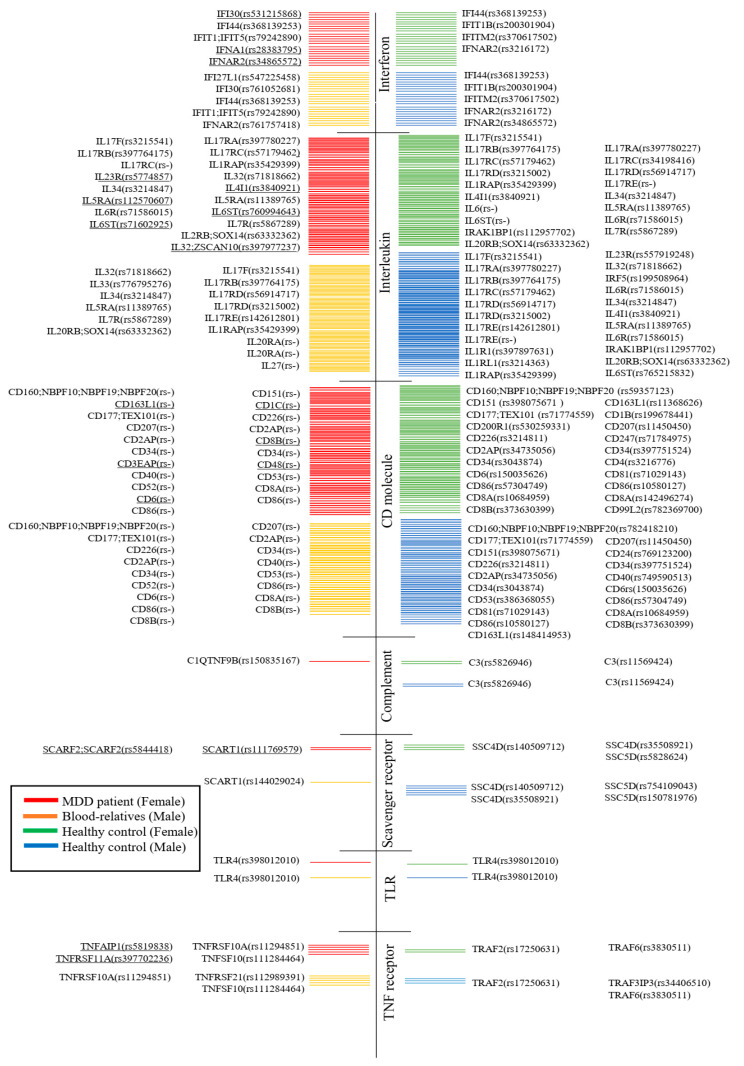
Whole-exome sequencing comparison of mutations of MDD patients, their relatives, and healthy controls. Underlines indicate them only being detected in MDD patients.

**Table 1 viruses-15-01898-t001:** Risk factors for MDD.

Factor	Group	N	MDD	Healthy	*p*-Value	Odds Ratio
Sex	Male	18	6 (33%)	12 (67%)	0.135	0.453 (0.158–1.300)
	Female	101	53 (52%)	48 (48%)		
Age		119	31 ± 6.4	22 ± 1.7	0.000	NA
Education level	No–High Vocational Certificate	23	16 (70%)	7 (30%)	0.033	2.817 (1.063–7.469)
	>High Vocational Certificate	96	43 (45%)	53 (55%)		
Income/month ($)	300–1000	67	41 (61%)	26 (39%)	0.004	2.979 (1.402–6.328)
	No–299	34	18 (53%)	16 (47%)		
Familial relationship	Quarrel	15	15 (100%)	0 (0%)	<0.001	NA
	No Quarrel	104	44 (42%)	60 (58%)		
BMI	>25	41	26 (63%)	15 (37%)	0.029	2.364 (1.085–5.147)
	<25	78	33 (42%)	45 (58%)		
Five main food groups	No	58	33 (57%)	25 (43%)	0.120	1.777 (0.859–3.674)
	Yes	61	26 (43%)	35 (57%)		
Exercise	No	59	37 (63%)	22 (37%)	0.004	2.905 (1.380–6.115)
	1–7 times/week	60	22 (37%)	38 (63%)		
Fresh fruit consumption (per week)	No–1 to 2 times	67	34 (51%)	33 (49%)	0.773	1.113 (0.539–2.297)
	3–7 times	52	25 (48%)	27 (52%)		
Vegetable consumption (per week)	No–1 to 2 times	40	20 (50%)	20 (50%)	0.948	1.026 (0.479–2.195)
	3–7 times	79	39 (49%)	40 (51%)		
High-fat food consumption (per week)	No–1 to 2 times	68	30 (44%)	38 (56%)	0.169	0.599 (0.288–1.246)
	3–7 times	51	29 (57%)	22 (43%)		
Fermented food consumption (per week)	3–7 times	59	10 (17%)	49 (83%)	0.014	5.918 (1.237–28.308)
	No–1 to 2 times	60	2 (3%)	58 (97%)		
Cleaning of water for consumption	No	36	13 (36%)	23 (64%)	0.053	0.455 (0.203–1.018)
	Yes	83	46 (55%)	37 (45%)		
Tap water used for brushing teeth	Yes	101	56 (55%)	45 (45%)	0.002	6.222 (1.695–22.836)
	No	18	3 (17%)	15 (83%)		
Alcohol consumption	Yes	85	41 (48%)	44 (52%)	0.643	0.828 (0.373–1.837)
	No	34	18 (53%)	16 (47%)		
Secondhand smoke exposure	Yes	46	33 (72%)	13 (28%)	<0.001	4.589 (2.060–10.221)
	No	73	26 (36%)	47 (64%)		

**Table 2 viruses-15-01898-t002:** Association of HHV-6 with MDD patients, relatives, and healthy participants.

Group	HHV-6 Positive	HHV-6 Negative	Total
MDD patients	15 (25.42%)	44 (74.58%)	59
Healthy subjects	51 (14.17%)	309 (85.83%)	360
Blood relatives of patients	17 (47.22%)	19 (52.78%)	36
Non-blood relatives of patients	7 (43.75%)	9 (56.25%)	16
Total	90 (19.11%)	381 (80.89%)	471

**Table 3 viruses-15-01898-t003:** The association of *TNF-α* promotor status with MDD patients, relatives, and the healthy population.

Group	*TNF-α* (-308G/A)	*TNF-α* (-308G)	Total
MDD patients	3 (7.69%)	36 (92.31%)	39
Healthy participants	15 (6.82%)	205 (93.18%)	220
Blood relatives of patients	6 (30.00%)	14 (70.00%)	20
Non-blood relatives of patients	2 (14.29%)	12 (85.71%)	14
Total	26 (8.87%)	267 (91.13%)	293

**Table 4 viruses-15-01898-t004:** Prevalence of HHV-6 infection, the *TNF-α* (-308G/A) promotor mutation, and MDD patients in Asia.

Country	Sample	Prevalence of HHV-6				Prevalence of TNF-α (-308G/A)			Prevalence of Depression	
		Method	Disease	Normal	Ref.	Disease	Normal	Ref.	Review Papers	Global Report [66]
**Thailand**	Saliva	Nested-PCR	-	5.7–8.6%	[60]	G/G = 166 (83.0%) G/A = 31 (15.5%) A/A = 3 (1.5%)	G/G = 181 (92.1%) G/A = 19 (7.9%) A/A = 0 (0.0%)	[67,68]	13.5% [69]	3.7%
	PBMC	Nested-PCR	-	45.5–78.3%	[60]	G/G = 108 (89.3%) G/A = 11 (9.1%) A/A = 2 (1.7%)	G/G = 113 (86.9%) G/A = 15 (11.5%) A/A = 2 (1.5%)	[70]	23.5 % [71]	
**Japan**	PBMC	qPCR	10–27%	-	[72]	G/G = 441 (96%) G/A = 19 (4%) A/A = 1 (0%)	G/G = 454 (98%) G/A = 10 (2%) A/A = 1 (0)	[73]	28.2% [74]	2.7%
	Blood & serum	qPCR& FISH	1/12 (8.3%)	46/85 (54.1%)	[75]	G/G = 57 (96.6%) G/A = 2 (3.4%) A/A = 0 (0.0%)	G/G = 556 (97%) G/A = 18 (3.1%) A/A = 1 (0.2%)	[76]	17% [77]	
	Blood	qPCR	0.6%	0.2%	[78]	G/G = 97% G/A = 4% A/A = 1%	G/G = 95% G/A = 5% A/A = 0%	[79]	3.1–6.6% [80,81]	
**Russia**	Blood	qPCR	1/124 (0.81%)	1/70 (1.7%)	[82]	G/G = 176 (74.6%) G/A = 53 (22.5%) A/A = 7 (3.0%)	G/G = 242 (79.9%) G/A = 55 (18.2%) A/A = 6 (2.0%)	[83]	20.7% [84]	3.9%
**Qatar**	Serum	ELISA	-	71.7%	[85]	G/G = 26 (70.3%) G/A = 10 (27.0%) A/A = 1 (2.7%)	G/G = 12 (52.2%) G/A = 11 (47.8%) A/A = 0	[86]	48% [87]	4.9%
**Vietnam**	Blood	qPCR	12.6%	-	[88]	G = 182 (87%) A = 28 (13%) G/G = 79 (75%) G/A = 24 (23%) A/A = 2 (2%)	G = 378 (93%) A = 30 (7%) G/G = 174 (85%) G/A = 30 (15%) A/A = 0	[89]	15.2% [90]	2.8%
**China**	Blood	qPCR	40%	16.2%	[91]	G = 1083 (90%) A = 117 (10%) G/G = 488 (81%) G/A = 107 (18%) A/A = 5 (1%)	G = 1142 (95%) A = 58 (5%) G/G = 543 (91%) G/A = 56 (9%) A/A = 1 (0%)	[92]	16.3–18.7% [93]	-
	CSF	qPCR	23/405 (5.7%)	-	[94]	G/G = 284 (83.8%) G/A = 50 (14.7%) A/A = 5 (1.5%)	G/G = 171 (84.7%) G/A = 31 (15.3%) A/A = 0 (0.0%)	[95]	2.0–2.5% [80,96]	-
**Hong Kong**	Blood	qPCR	4/10 (40%)	2/10 (20%)	[97]	G/G = 88 (90%) G/A = 10 (10%) A/A = 0 (0%)	G/G = 90 (93%) G/A = 6 (6%) A/A = 0 (0.0%)	[98]	3.7% [99]	-
**Indonesia**	PBMC	qPCR	15/85 (17.6%)	3/85 (3.5%)	[100]	G/G = 78% G/A = 22% A/A = 0%	G/G = 92% G/A = 8% A/A = 0%	[101]	16.3%[102]	2.6%
**Korea**	PBMC	PCR	7/34 (20.6%)	0/20 (0%)	[65]	G/G = 110 (92%) G/A = 9 (8%) A/A = 0 (0%)	G/G = 115 (85%) G/A = 20 (15%) A/A = 0 (0%)	[103]	10.1–14.3% [104,105]	-
**India**	Saliva	PCR	40%	35%	[106]	G = 429 (80%) A = 109 (20%) G/G = 169 (63%) G/A = 91 (34%) A/A = 9 (3%)	G = 502 (92%) A = 42 (8%) G/G = 230 (85%) G/A = 42 (15%) A/A = 0 (0%)	[92]	6–9% [81,107,108]	3.7%
**Taiwan**	PBMC	qPCR	43.5%	-	[109]	G/G = 121 (85%) G/A = 22 (15%) A/A = 0 (0%)	G/G = 31 (53%) G/A = 25 (42%) A/A = 3 (5%)	[103]	1.3–2.4% [110]	3.6%
**Malaysia**	Oral cell	PCR	19/24 (79%)	0/7 (0%)	[111]	G/G = 85 (86.7%) G/A = 11 (11.2%) A/A = 2 (2%)	G/G = 49 (86%) G/A = 7 (12.3%) A/A = 1 (1.8%)	[112]	11% [113]	3.9%

**Table 5 viruses-15-01898-t005:** Whole-exome sequencing comparison of mutations in MDD patients, their relatives, and healthy controls.

Gene	Disease	MDD Mutation		Rs	Healthy Mutation		Rs
		Patient	Relative		Female	Male	
ATP Binding Cassette Transporter G1 (ABCG1) [114]	Major depressive disorder (MDD)	/	-	rs-	-	-	-
Acetylserotonin O-Methyltransferase (ASMTL) [115]	Recurrent depressive disorder	/	/	rs13329185	-	-	-
Calcium Voltage-Gated Channel Subunit Alpha1 F (CACNA1F) [116]	Psychiatric disorders	/	-	rs371501501	-	-	-
Calcium Voltage-Gated Channel Subunit Alpha1 I (CACNA1I) [116]	Psychiatric disorders	/	-	rs-	-	/	rs-
Calcium Voltage-Gated Channel Auxiliary Subunit Alpha 2 Delta 2 (CACNA2D2) [117]	Schizophrenia	-	/	rs-	-	-	-
Cytochrome C Oxidase Subunit 7A1 (COX7A1) [118]	Neurodegeneration	/		rs112834485	/	-	rs755756126
Cytochrome C Oxidase Subunit 7A2 (COX7A2) [118]	Neurodegeneration	/	/	rs554727448	-	-	-
Glutamate Ionotropic Receptor Delta Type Subunit 2 (GRID2) [119]	Depolarization of neurons	/	-	rs-	-	-	-
5-Hydroxytryptamine Receptor 3E (HTR3E) [120]	Schizophrenia	/	/	rs150341032	-	/	rs187832026
SH3 And Multiple Ankyrin Repeat Domains 2 (SHANK2) [121]	Neurodevelopmental disorder	/	/	rs35132270	-	-	-
Tryptophan Hydroxylase-2 (TPH2) [122]	MDD	/	-	rs397897324	-	-	-

## Data Availability

Not applicable.

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
