# Peer review of "The Association of HHV-6 and the TNF-α (-308G/A) Promotor with Major Depressive Disorder Patients and Healthy Controls in Thailand"

_viruses, 2023, doi:10.3390/v15091898_

Round 1

Reviewer 1 Report

Dear Authors,

The subject of the manuscript titled "The Association of HHV-6 and the TNF-α (-308G>A) Promoter with Major Depressive Disorder Patients and Healthy Controls in Thailand" is quite relevant and certainly arouses interest.

However, to meet the level of the journal, I recommend significant revisions starting from the abstract and introduction and ending with the conclusion.

Abstract and Introduction: Please consider restructuring the abstract and introduction to follow a clear and logical sequence. Include the general problem, specific problem, research gaps, purpose of the study, and the methods employed to address the research goals. Highlight the significance of investigating the correlation between TNF-α gene variation and major depressive disorder (MDD) in the Thai population to provide context for the readers.

Materials and Methods: To enhance clarity, consider introducing a flow chart to illustrate the study groups and sampling procedures (Thus, in lines 130-132 the authors postulate that 1) “A total of 471 oral buccal cell samples were collected from 360 healthy individuals, 59 MDD patients, 36 blood relatives of MDD patients, and 16 non-blood relatives of MDD patients.” With that, in lines 142-143 one can read that “randomized sample of 60 healthy individuals and 59 MDD patients was used to 142 confirm the most common risk factors of MDD” and in lines 227-229 one can “Oral buccal cells were collected from 471 donors from the Thai population. HHV-6 was detected by qPCR.” Please make it clear for readers.). Additionally, provide explicit information about the number of samples collected from each group to avoid confusion. Include details on ethical approval and patient consent in accordance with STROBE recommendations for studies conducted on humans.

Results and Discussion: Rework the results and discussion sections to emphasize the findings, which are of significant interest to the scientific community. Focus on the implications of the observed association between TNF-α gene variation and MDD, considering its influence on TNF-α levels and potential impact on immunological homeostasis. Address the lack of information on the prevalence of the -308G>A allele in people with depression and its relationship with viral load to enhance the understanding of MDD's complex etiology.

Conclusion: Revise the conclusion to highlight the novel insights derived from the study and their implications for future research and clinical applications. Emphasize the relevance of investigating TNF-α gene variation in the context of MDD among the Thai population, encouraging further investigation into this intriguing association.

Ethical Considerations: Provide the necessary data on ethical approval and patient consent to align with ethical standards for studies conducted on humans. In addition, consider adopting the STROBE recommendations to enhance the presentation of the study, particularly regarding the formulation of the introduction, highlighting the general problem, specific problem, research gaps, purpose of the study, and methods used to address the research goals.

Clearly state any limitations of the study and suggest directions for future research.

Overall, your article presents an intriguing and relevant topic, but revisions are essential to improve clarity and adhere to journal standards. Emphasizing the intriguing findings will enhance the impact of the manuscript and align it with the level of the journal.

Thank you for considering these suggestions. I look forward to reviewing the revised version of your article.

The English language requires stylistic improvement to enhance readability, particularly in the Introduction and Discussion sections, where a series of loosely connected sentences are present.

Author Response

Respond to the reviewer according to the attach file

Reviewer 2 Report

The aims of the study by Sumala et al. Were to study the association between HHV6 reactivation, TNF-α promoter polymorphism, and major depressive disorder (MDD) in the adult Thai population. Although the authors found a more frequent presence of HHV6 in buccal swabs from patients with MDD in comparison with healthy controls, the results obtained cannot provide a definite answer to this question. The problem is in serious mistakes in the study design: 1) MDD patients were enrolled from the general population based on the questionnaires, although the authors themselves dispute this method of MDD diagnosis (see Introduction, line 118-9). No data are available on disease treatment or on the exclusion of other chronic or acute inflammatory states. 2) The control group was randomly selected, but it would be matched minimally by age and sex. 3) Buccal cells isolated from self-sampled swabs are not optimal for the measurement of HHV6 reactivation because the samples are not standardized and may contain variable amounts of biological material. Saliva or blood samples were more appropriate (see Tabel 4 in the paper). 4) The number of samples tested for cytokine levels was too small to obtain statistically significant results. No conclusions can be drawn from the NGS analysis of only four individuals (one patient and two controls). 5) HHV6 infection, instead of HHV6 reactivation, is incorrectly used in many places, giving the impression that the authors wrongly understand the HHV6 life cycle and epidemiological behavior.

Other comments on the paper

The introduction is not well arranged: it contains data that are irrelevant with respect to the problem solved (lines 46-51). However, there is a lack of basic data on the HHV6 life cycle and epidemiology, particularly in relation to its association with neurological diseases. Some statements in the Introduction section are incorrect (....HHV6 is associated with increase of neurological disease... HHV6 is biomarker for fatigue and chronic fatigue syndrome...., and other ( line 88, line 95, line 111).

Material and methods: The virus quantitation by number of DNA copies per 5 ul of DNA isolate does not possess adequate information. The PCR results should be related to the copy number of the control cellular gene or to the amount of genomic DNA. Which types of samples were used in cytokine ELISAs? In which dilution?

Results: line 219-sex was not significantly associated with MDD. On the other hand, smoking was omitted ( see Table 1). Table 1 – What does mean p= 0,000?. Which congenital disease?

Fig. 2 – statistical analysis is lacking

Par. 3.4. Viral loads in copy numbers per µL DNA were not comparable between the groups (see comment for M + M).

Fig. 3 Statistical evaluation of cytokine levels associated with TNFα polymorphisms depicted in Fig. 3A,B, and C is not credible because of the small number of samples measured and too low levels found (assay ranges for IL-10 and TNFα start from 8 and 15 pg/mL. Samples with concentrations under the assay range were negative in the assay and should not be quantitated). Fig 3D does not contain any data on the TNFα promoter status.

The discussion lacks a confrontation of the results with those presented in the literature ( see Table 4). Instead, it presents various hypotheses that are not based on the results. Owing to the high prevalence of HHV6 infection in the population and its ability to reactivate under any inflammatory or immunosuppressive conditions, it is difficult to discern whether an association between HHV6 reactivation and a disease is causal or bystender.

Conclusions are not well founded.

Author Response

(The authors gave the same response as above.)

Reviewer 3 Report

I have read this review throughout, which mainly discusses the role of HHV-6 and TNF- α(-308 G>A) in the risk of MDD. The entry point of this manuscript is excellent, but there is still insufficient understanding of research. The following modifications are required:

1. The author should ensure whether there are any factors that affect the occurrence of MDD in non HHV-6 infected individuals.

2. The author can also analyze and compare the incidence of MDD between HHV-6 infected individuals and non HHV-6 infected individuals.

3. Due to the lack of experimental validation, the author is unable to derive the conclusion that TNF- α (-308 G>A) promotes HHV-6 infection and MDD risk. Perhaps these phenomena are simply related and have no causal relationship.

Author Response

(The authors gave the same response as above.)

Reviewer 4 Report

Despite the interesting and intriguing topic raised by the authors in the proposed manuscript, its scientific value is rather low, as in most cases incorrect analysis of the data and thus the presentation and evaluation of the results is misleading to the reader.

If the authors want to publish the obtained results, it is necessary to significantly and correctly revise part of the statistical processing and correctly describe and interpret the new results obtained, only then the manuscript can be accepted for publication.

See comments and suggestion in the attachment.

Author Response

(The authors gave the same response as above.)

Round 2

Reviewer 1 Report

Dear authors, thank you for the conducted work. I do not have any additional comments

Author Response

Thank you for your suggestions and time!

Reviewer 2 Report

see attached

Author Response

Please see attached, thank you!

Reviewer 4 Report

Corrections were made and the manuscript improved in some way .

Author Response

(The authors gave the same response as above.)
